# The 'de novo' DNA methyltransferase Dnmt3b compensates the Dnmt1-deficient intestinal epithelium

**Ellen N Elliott**[1,2†‡], **Karyn L Sheaffer**[1,2†], **Klaus H Kaestner**[1,2*]

[1]Department of Genetics, Perelman School of Medicine, University of Pennsylvania, Philadelphia, United States; [2]Institute for Diabetes, Obesity and Metabolism, Perelman School of Medicine, University of Pennsylvania, Philadelphia, United States

**Abstract** *Dnmt1* is critical for immediate postnatal intestinal development, but is not required for the survival of the adult intestinal epithelium, the only rapidly dividing somatic tissue for which this has been shown. Acute *Dnmt1* deletion elicits dramatic hypomethylation and genomic instability. Recovery of DNA methylation state and intestinal health is dependent on the de novo methyltransferase Dnmt3b. Ablation of both *Dnmt1* and *Dnmt3b* in the intestinal epithelium is lethal, while deletion of either *Dnmt1* or *Dnmt3b* has no effect on survival. These results demonstrate that Dnmt1 and Dnmt3b cooperate to maintain DNA methylation and genomic integrity in the intestinal epithelium.

**\*For correspondence:** kaestner@mail.med.upenn.edu

†These authors contributed equally to this work

**Present address:** ‡The Jackson Laboratory for Genomic Medicine, Farmington, United States

**Competing interests:** The authors declare that no competing interests exist.

## Introduction

DNA methylation patterns are established by DNA methyltransferase enzymes (Dnmts), for which two categories have been defined based on in vitro assays. The 'de novo' methyltransferases, Dnmt3a and Dnm3b, establish novel patterns of DNA methylation, and prefer to bind unmethylated DNA in vitro (*Okano et al., 1998*). The 'maintenance' methyltransferase, Dnmt1, has a high affinity for hemi-methylated DNA in vitro, and preserves DNA methylation in replicating cells (*Bestor, 1992*; *Leonhardt et al., 1992*). Dnmts are crucial for embryonic development in mice, as mice null for *Dnmt1* or *Dnmt3b* arrest at mid-gestation, and *Dnmt3a* null mice die in the first few weeks of life (*Li et al., 1992*; *Okano et al., 1999*). Although DNA methylation is not necessary for murine embryonic stem (ES) cell growth, the differentiation of *Dnmt1*-hypomorphic and *Dnmt3a*; *Dnmt3b*-mutant ES cells is severely impaired (*Chen et al., 2003*; *Jackson et al., 2004*; *Lei et al., 1996*; *Tsumura et al., 2006*). These results indicate an important role for DNA methylation and Dnmts in directing cell differentiation processes.

One of the primary consequences of hypomethylation is increased DNA damage and genomic instability. Global hypomethylation in mice results in chromosome duplications and invasive T-cell lymphomas at four months of age (*Gaudet et al., 2003*), and in mouse ES cells, loss of *Dnmt1* also causes global hypomethylation and increased mutation rates (*Chen et al., 1998*). In the HCT116 colorectal cancer cell line, ablation of the catalytically active DNMT1 results in cell cycle arrest and apoptosis due to increased chromosomal instability (*Chen et al., 2007*; *Spada et al., 2007*). In mouse embryonic fibroblasts, ablation of either *Dnmt1* (*Jackson-Grusby et al., 2001*) or *Dnmt3b* (*Dodge et al., 2005*) causes gradual hypomethylation, deregulated gene expression, and cell death. Dnmt1 and DNA methylation are also required for viability in most proliferating somatic cell populations, including human skin cells (*Sen et al., 2010*), mouse embryonic fibroblasts (*Jackson-*

**eLife digest** Genes in a cell can be switched on or off at different times depending on the cell's requirements. Small chemical groups can be attached to the gene's DNA, which dictates whether it is activated or inactivated. For example, methyl groups can be attached to DNA by enzymes known as DNA methytransferases in a process called DNA methylation.

In mammals, the pattern of DNA methylation changes in a highly regulated manner as the embryo develops. Mice that lack enzymes called DNA methytransferase 1 or DNA methytransferase 3b (shortened to Dnmt1 or Dnmt3b) die before they are born. It is also widely believed that Dnmt1 is needed to preserve DNA methylation patterns in dividing cells, and this enzyme is often called a 'maintenance' methyltransferase because it maintains the DNA methylation pattern in newly formed cells.

Epithelial cells that form an animal's intestine are constantly being produced from dividing cells found in folds of the intestine called crypts. These cells are among the most rapidly dividing cells in animals, and are replaced every three to five days throughout adult life. Recently in 2015, researchers deleted the gene for Dnmt1 in intestinal cells in adult mice. This caused a reduction in DNA methylation and led to abnormal gene activation. However, the crypt cells were still able to form new cells to renew the intestine.

Elliott, Sheaffer and Kaestner – who were all involved in the previous study – have now explored why intestinal cells in adult mice can survive without Dnmt1. First, the experiments showed that the mutant mice recovered their normal levels DNA methylation within two months of the gene deletion. Further analysis uncovered that this recovery was due to the fact that Dnmt3b became activated in Dnmt1-lacking cells, and then re-methylated the DNA. Deleting the genes for both Dnmt1 and Dnmt3b led to loss of DNA methylation and many of the adult mice died prematurely.

Previously, it was thought that Dnmt3b only acted to establish new DNA methylation patterns, but these latest findings suggest that this enzyme can act a 'maintenance' methyltransferase as well. Together the findings also reveal that Dnmt1 and Dnmt3b cooperate to maintain methylation in the epithelial cells in the intestines of adult mice. Further work could next investigate if this is also the case for other tissues in the body.

Grusby et al., 2001), and neuronal (*Fan et al., 2001*) and pancreatic (*Georgia et al., 2013*) progenitor cells.

Interestingly, Dnmt1 is not required for adult intestinal stem cell survival (*Sheaffer et al., 2014*). The mature intestinal epithelium is a single cell layer lining the lumen of the intestine, structured into finger-like protrusions, designated 'villi', and invaginations into the underlying mesenchymal tissue, termed 'crypts.' Intestinal stem cells are located in the crypt and respond to multiple signaling pathways that control proliferation and differentiation (*Elliott and Kaestner, 2015*). Stem cells give rise to rapidly dividing transit-amplifying cells, which move in ordered cohorts up the crypt-villus axis. As cells migrate out the crypt, they differentiate into one of several distinct cell lineages, a process that is largely dependent on levels of Notch signaling. Loss of Dnmt1 in the adult mouse intestinal epithelium causes hypomethylation of regulatory regions associated with several intestinal stem cell genes, resulting in inappropriate gene expression during differentiation, and expansion of the crypt zone (*Sheaffer et al., 2014*). In contrast, ablation of *Dnmt1* during intestinal crypt development causes hypomethylation, DNA damage, and apoptosis of epithelial cells, resulting in increased perinatal lethality (*Elliott et al., 2015*). Previous studies did not investigate the requirement for Dnmt1 in maintaining global DNA methylation or preserving genomic stability in the mature intestine. Thus, the mechanism behind preservation of the *Dnmt1*-mutant adult intestinal epithelium, the only rapidly dividing somatic tissue known to survive without Dnmt1, is not known.

To determine the mechanism underlying *Dnmt1* mutant intestinal survival, we employed tissue-specific, inducible mouse models and analyzed the effects immediately after *Dnmt1* deletion in the adult intestinal epithelium. Ablation of *Dnmt1* caused an acute phenotype characterized by weight loss, global DNA hypomethylation, genome instability, and apoptosis. Strikingly, animals returned to baseline DNA methylation levels within two months of *Dnmt1* deletion, indicating recovery by a de

novo methyltransferase. We demonstrate that the de novo methyltransferase Dnmt3b is upregulated following loss of Dnmt1, and essential for epithelial survival in the Dnmt1 mutant intestine. Our results implicate a role for DNA methylation, maintained by both Dnmt1 and Dnmt3b, in protecting genomic stability in intestinal epithelial homeostasis. These data are the first to show that Dnmt3b can function in maintenance DNA methylation in vivo.

## Results

### *Dnmt1* ablation results causes weight loss, hypomethylation, and genomic instability

To determine the primary effects of Dnmt1 deletion in the adult intestinal epithelium, we employed an inducible, intestinal epithelial-specific gene ablation model. The $Dnmt1^{loxP/loxP}$;Villin-CreERT2 mice (Dnmt1 mutants) and their $Dnmt1^{loxP/loxP}$ siblings (controls) were tamoxifen-treated at four weeks of age to induce Cre recombinase activity (*El Marjou et al., 2004*; *Jackson-Grusby et al., 2001*). Although Dnmt1 mutants lost a significant amount of weight in the two weeks following Cre induction, mice recovered by day 17 post-tamoxifen treatment, and survived at rates identical to controls (*Figure 1A–B*). To elucidate the mechanism underlying acute weight loss, we isolated Dnmt1 mutant and control intestines one week following tamoxifen treatment.

One-week post-tamoxifen treatment, Dnmt1 mutants exhibited multiple abnormalities in intestinal epithelial morphology, with partial loss of epithelial integrity and a high frequency of crypt fission (*Figure 1C–D*). Dnmt1 loss was confirmed on the protein level by immunohistochemistry (*Figure 1E–F*), and the Dnmt1 mutant epithelium displayed a slight expansion of the proliferative crypt zone (*Figure 1G–H*), as reported previously (*Sheaffer et al., 2014*). However, Dnmt1 mutant epithelia also exhibited regions that lacked crypts and/or villi, juxtaposed with hyperplastic crypts that replaced damaged tissue (*Figure 1D*). Since Dnmt1 ablation results in increased double stranded breaks and apoptosis in the neonatal intestine (*Elliott and Kaestner, 2015*), we investigated if the Dnmt1 deficient adult intestine also displayed altered genomic stability. As a general marker of chromosomal instability, we stained for γH2AX, which labels DNA double strand break foci, and is an indicator of the DNA damage response. We observed an increase in γH2AX foci in the crypts of Dnmt1 mutants compared to controls, which displayed minimal DNA double strand breaks (*Figure 1G–H*). We also performed TUNEL staining to identify apoptotic nuclei, and found that the Dnmt1 mutant intestine exhibited increased crypt cell apoptosis relative to the control intestinal epithelium, which did not contain any apoptotic nuclei (*Figure 1I–J*).

To determine global DNA methylation levels, we isolated mutant and control intestinal crypt epithelium by laser capture microdissection, and performed targeted bisulfite-sequencing of the repetitive *LINE1* loci. *LINE1* retrotransposons account for approximately 20% of the mouse genome (*Waterston et al., 2002*), and are a representative of genome-wide DNA methylation levels (*Lane et al., 2003*; *Yang et al., 2004*). The *LINE1* repeats were significantly demethylated in Dnmt1 mutant crypts, with methylation reduced to approximately 50% of controls at each CpG analyzed (*Figure 1K*). We also performed bisulfite sequencing of the *H19* imprinting control region to analyze maintenance methylation, and found that the region was slightly demethylated relative to controls, although this difference was only significant when comparing the methylation of the entire sequenced region (*Figure 1L*). Overall, these results suggest a phenotype in which loss of Dnmt1 results in hypomethylation of LINE1 repeats and the H19 imprinting control region, increased DNA damage, and apoptosis.

### Dnmt1-deficient intestinal epithelium recovers within two months of tamoxifen treatment

Interestingly, $Dnmt1^{loxP/loxP}$;Villin-CreERT2 mutant mice survive at rates comparable to controls (*Figure 1B*), indicating that Dnmt1 is not required for continued intestinal maintenance in the adult mouse. To determine the long-term effects of *Dnmt1* ablation, we harvested Dnmt1 mutant and sibling control intestines two months following tamoxifen injection. We confirmed that Dnmt1 deletion had been maintained in mutant epithelia (*Figure 2C–D*), but found that Dnmt1 mutant intestinal epithelial morphology (*Figure 1A–B*) and proliferation (*Figure 1E–F*) were comparable to controls. In addition, levels of DNA damage and apoptosis, as indicated by γH2AX and TUNEL staining,

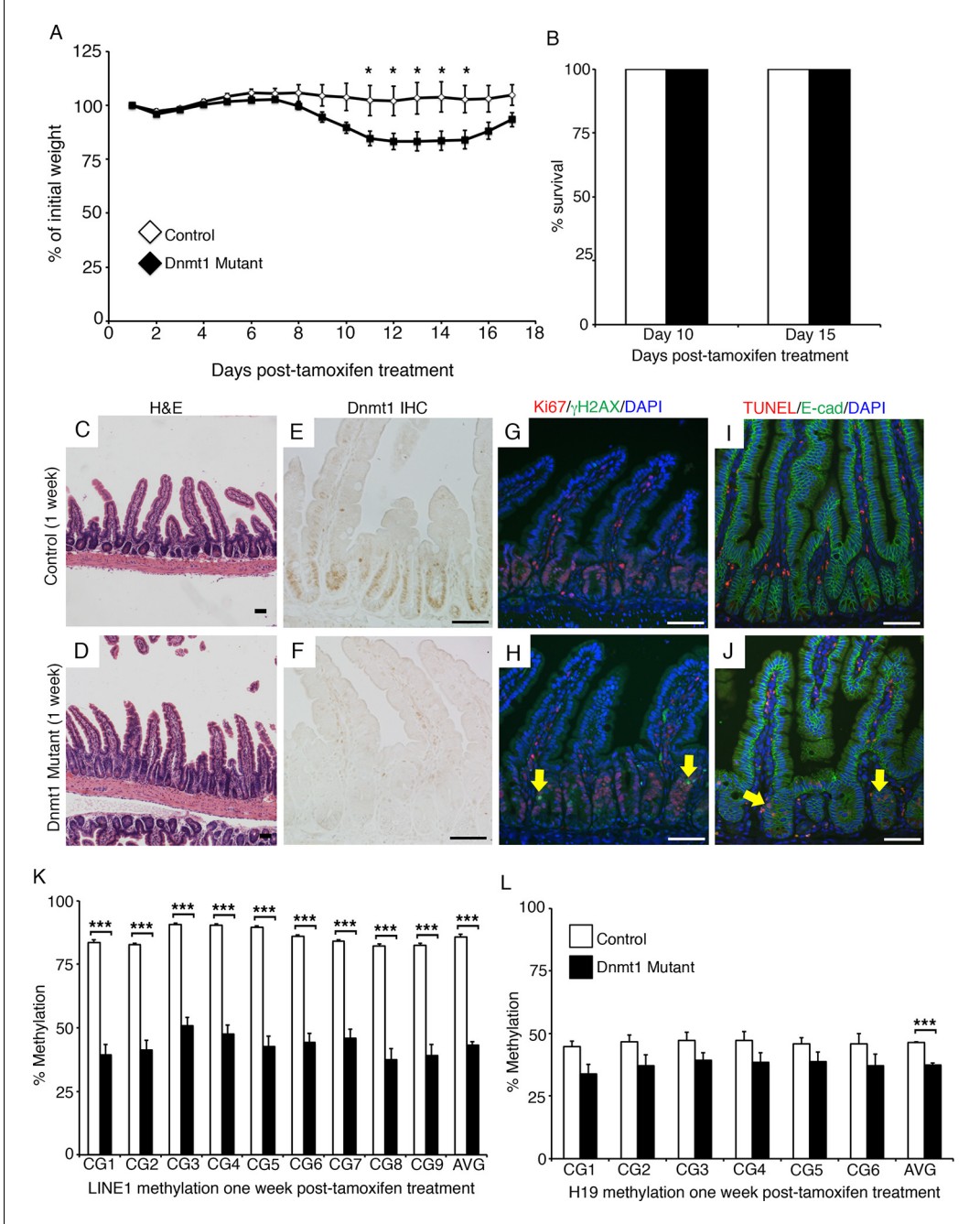

**Figure 1.** *Dnmt1* ablation results in genomic instability and apoptosis one week following tamoxifen treatment. (A, B) *Dnmt1^{loxP/loxP}*(control, n=5) and *Dnmt1^{loxP/loxP};Villin-CreERT2* (Dnmt1 mutant, n=9) mice were tamoxifen treated at four weeks of age, and weighed every day over a 17-day period. Dnmt1 mutants lost a significant amount of weight by day 11, but recovered to near-starting weight by day 16. *p<0.05, Student's *t*-test. (B) All *Dnmt1* mutants survive the 17-day time-course described in (A), similar to controls. (C–D) Hematoxylin and Eosin staining of control and Dnmt1 mutant intestines. One week post-ablation, Dnmt1 mutants exhibit loss of crypt integrity, vacuolization of the epithelium, and an increase in crypt fission (D) compared to controls (C). (E–F) Immunohistochemistry confirms loss of Dnmt1 protein in mutants one week following tamoxifen treatment (F) relative to control intestine (E). (G-H) Immunofluorescent staining for Ki67 (red), which marks proliferating cells, and γH2AX (green), which marks DNA double-strand breaks as a marker of chromosomal instability. One week following Dnmt1 ablation, mutant crypts display increased levels of γH2AX foci (H, yellow arrows) relative to controls (G). Dnmt1 mutants also display slightly enlarged crypts (Ki67 in H versus G), as described previously (*Sheaffer et al., 2014*). (I–J) Immunofluorescent TUNEL staining (red), which marks apoptotic nuclei, and E-cadherin (green), to outline the intestinal epithelium. One week after tamoxifen treatment, Dnmt1 mutants display increased crypt cell apoptosis (J, yellow arrows) compared to controls (I). (K–L) Crypts were isolated from paraffin-embedded tissue by laser capture microdissection, and the methylation levels of *LINE1* loci and the imprinting control region of *H19* were determined by targeted bisulfite sequencing. One week after tamoxifen treatment, methylation of LINE1 (K) and *H19* (L) are significantly

*Figure 1 continued on next page*

*Figure 1 continued*

decreased in Dnmt1 mutants compared to controls (n=4 per genotype). \*\*\*p<0.001, \*p<0.05, Student's *t*-test. For data and *p* values, refer to
**Figure 1—source data 1**. Error bars represent S.E.M. Scale bars are 50 µm. For all staining, n=3 biological replicates.
The following source data is available for figure 1:

**Source data 1.** Contains mouse weight/survival data in **Figure 1A–B**, targeted bisulfite sequencing data in **Figure 1K–L**.

respectively, were similar to control mice (**Figure 2E–H**). We isolated crypt cells from two-month Dnmt1 mutants and controls by laser-capture microdissection, and performed targeted bisulfite sequencing for *LINE1* and *H19*, as described above. LINE1 methylation levels remained significantly decreased compared to controls (**Figure 2I**), but demethylation was not as severe compared to Dnmt1 mutants at one-week post-tamoxifen treatment (compare **Figure 2I** to 1K). Strikingly, methylation at the *H19* imprinting control region had been fully restored (**Figure 2J**). These results implicate a mechanism that compensates for loss of Dnmt1 and leads to recovery of intestinal epithelial DNA methylation and genomic integrity.

## The de novo methyltransferase Dnm3b, but not Dnmt3a, is upregulated following Dnmt1 deletion

We surmised that the de novo methyltransferases might compensate for loss of Dnmt1, and performed qRT-PCR and immunofluorescent staining for both Dnmt3a and Dnmt3b in mutant and control intestines harvested one week following tamoxifen treatment. We confirmed loss of *Dnmt1* mRNA in mutant crypts by qRT-PCR, but observed no changes in *Dnmt3a* transcript or protein expression (**Figure 3A–C**). In addition, simultaneous loss of both Dnmt1 and Dnmt3a did not alter or exaggerate the Dnmt1 mutant phenotype, or result in decreased viability (**Figure 3—figure supplement 1**). *Dnmt1^loxP/loxP^;Dnmt3a^loxP/loxP^;Villin-CreERT2* mice (Dnmt1;Dnmt3a mutants) displayed acute changes in cell death, and *LINE1* and *H19* methylation, identical to those seen in *Dnmt1* mutants (**Figure 3—figure supplements 1G-H** and **2A–B**). Furthermore, *Dnmt1;Dnmt3a* mutants do not exhibit increased γH2AX foci compared to single Dnmt1 deficient mice (**Figure 3—figure supplement 1E–F**). Combined with its unchanged protein and mRNA expression, we concluded that Dnmt3a is not required for the recovery of *Dnmt1* mutant epithelia.

In contrast, *Dnmt3b* mRNA expression was significantly increased in the *Dnmt1* mutant crypts compared to control crypt cells (**Figure 3A**). In agreement with our qRT-PCR data, we discovered an increase in Dnmt3b protein levels in the mutant intestinal epithelium while expression levels in the lamina propria were unchanged (**Figure 3D–E**), suggesting that *Dnmt3b* is upregulated within one week of *Dnmt1* ablation. Overall, these data suggest a mechanism in which *Dnmt3b* is activated to counteract the loss of *Dnmt1* in the intestinal epithelium. We next aimed to test this proposed compensation mechanism using mouse genetics.

## Dnmt3b is required for survival of intestinal epithelial-specific Dnmt1 mutant mice

To directly test the requirement for *Dnmt3b* in maintaining DNA methylation in the *Dnmt1* mutant intestinal epithelium, we bred the *Dnmt3b^loxP/loxP^* allele onto the mutant genotype (**Lin et al., 2006**), producing *Dnmt1^loxP/loxP^;Dnmt3b^loxP/loxP^;Villin-CreERT2*, along with *Dnmt1^loxP/loxP^;Dnmt3b^loxP/loxP^* siblings as controls. To assess the overall requirement for *Dnmt3b* in *Dnmt1* mutant survival, we injected tamoxifen into groups of *Dnmt1^loxP/loxP^;Dnmt3b^loxP/loxP^;Villin-Cre-ERT2* mutants (*Dnmt1;Dnmt3b* mutants) and littermate controls at four weeks of age, and weighed mice each day following CreERT2 induction. 60% of *Dnmt1;Dnmt3b* mutant mice (n=10) became severely morbid within two weeks following tamoxifen administration and had to be euthanized (**Figure 4A–C**). *Dnmt1;Dnmt3b* mutants lost significantly more weight compared to *Dnmt1* mutants, which contributed to the increased lethality observed in double mutant mice (**Figure 4B–C**). In contrast, 94% of *Dnmt1* mutant mice (n=16) survived to 17 days following tamoxifen injection, confirming that loss of *Dnmt1* alone is non-lethal in the mature intestinal epithelium (**Figure 4A**). *Dnmt1;Dnmt3b* mutant mice that survived contained intestinal epithelium positive for *Dnmt3b* (**Figure 4—figure supplement 1A–B**) resulting from inefficient Cre-mediated gene ablation followed by expansion of "escaper" crypts, consistent

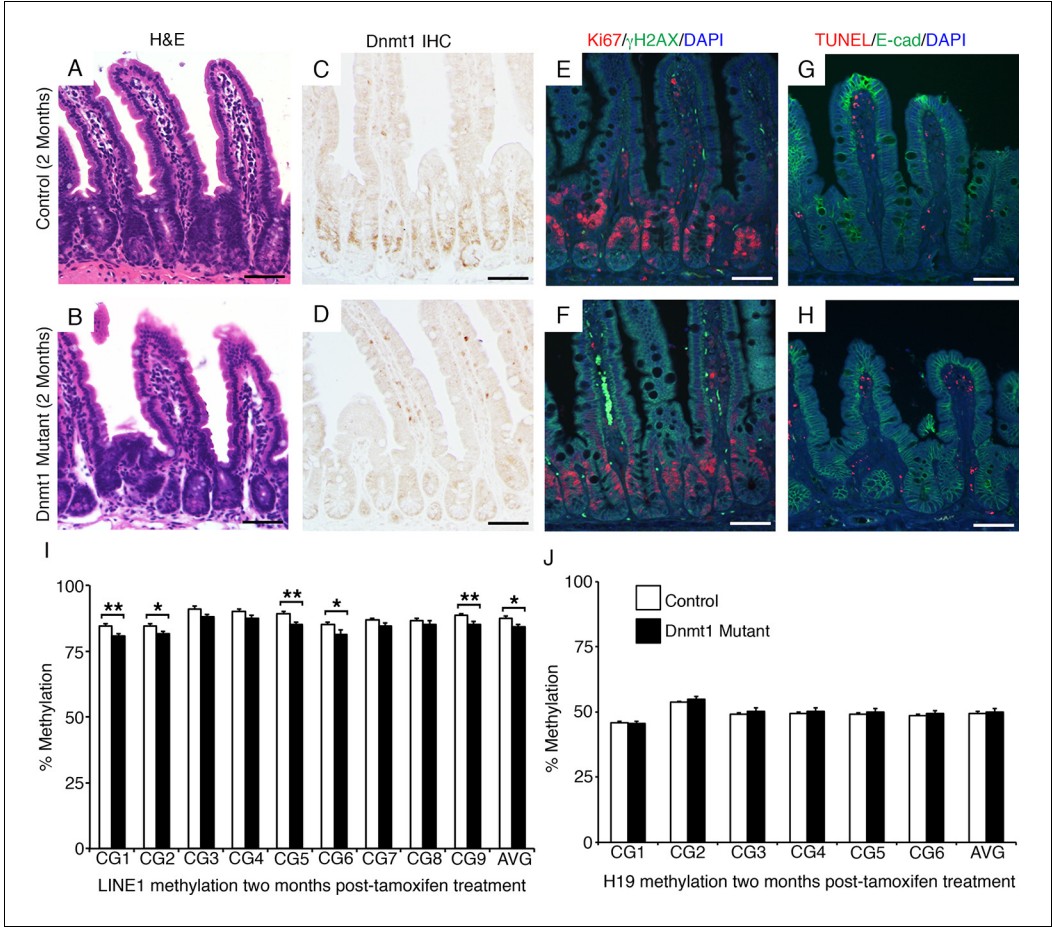

**Figure 2.** The Dnmt1 mutant intestinal epithelium recovers with time. $Dnmt1^{loxP/loxP}$ (control) and $Dnmt1^{loxP/loxP}$;$Villin$-$CreERT2$ (Dnmt1 mutant) mice were tamoxifen treated at four weeks of age, and intestines were harvested two months later for immunostaining and DNA methylation analysis. (**A–B**) Hematoxylin and eosin staining revealed that two months following tamoxifen treatment, the Dnmt1 deficient epithelium appears similar to controls (**B** versus **A**). (**C–D**) Epithelial Dnmt1 deletion is maintained in Dnmt1 mutants two months after tamoxifen injections (**D** versus **C**). (**E–F**) Immunofluorescent staining for Ki67 (red), which marks proliferating cells, and γH2AX (green), which marks DNA double-strand breaks as a marker of chromosomal instability. By two months post-Dnmt1 deletion, the mutant epithelium has returned to baseline levels of DNA damage (**F** versus **E** control). (**G–H**) TUNEL staining (red), which marks apoptotic nuclei, and immunostaining for E-cadherin (green), to outline the intestinal epithelium. Two months following tamoxifen injection, Dnmt1 mutants appear similar to controls and display no apoptosis in the epithelium (**H** versus **G**, respectively). (**I,J**) Crypts were isolated from paraffin-embedded tissue by laser capture microdissection, and the methylation levels of $LINE1$ loci and the imprinting control region of $H19$ were determined by targeted bisulfite sequencing. Two months following tamoxifen injection, Dnmt1 mutants have mostly regained methylation at both the LINE1 (**E**) and $H19$ (**F**) loci, and are comparable to controls (n=5–6 per genotype). However, the slight demethylation across the entire LINE1 loci is significantly decreased compared to controls. For data and $p$ values per CpG, refer to **Figure 2—source data 1**. Error bars represent S.E.M. Scale bars are 50 µm. For all staining, n=3 biological replicates.

The following source data is available for figure 2:

**Source data 1.** Contains targeted bisulfite sequencing data presented in **Figure 2I–J**.

with our hypothesis that $Dnmt3b$ is required to preserve epithelial integrity in the absence of $Dnmt1$.

Next, we isolated small intestines from $Dnmt1;Dnmt3b$ mutants and sibling controls one week after tamoxifen administration for DNA methylation and immunostaining analysis. We employed laser-capture microdissection to isolate one-week $Dnmt1;Dnmt3b$ mutant and control crypt cells for DNA methylation analysis. As expected, $Dnmt1;Dnmt3b$ mutants were significantly demethylated at the LINE1 loci compared to controls (**Figure 5A**). Although the $Dnmt1;Dnmt3b$ mutants displayed decreased methylation levels relative to Dnmt1 mutants, this difference was not statistically

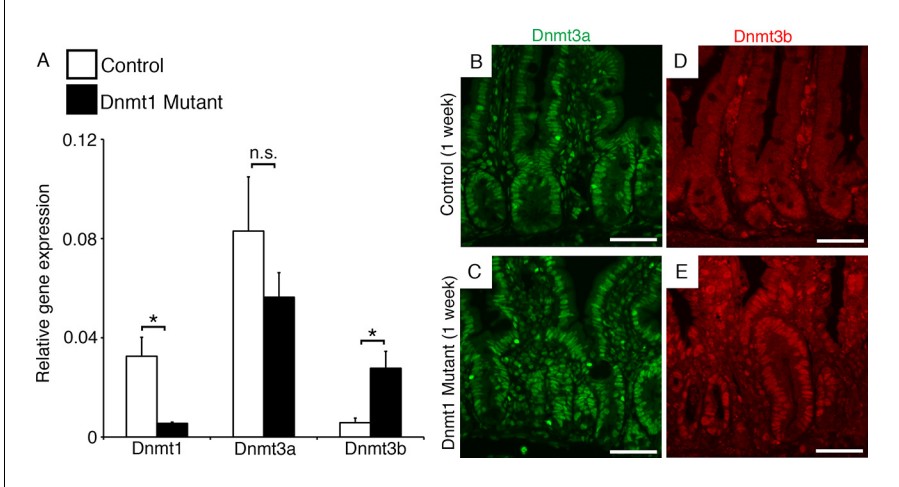

**Figure 3.** *Dnmt3b* is upregulated following *Dnmt1* ablation. *Dnmt1$^{loxP/loxP}$*(control) and *Dnmt1$^{loxP/loxP}$;Villin-CreERT2* (Dnmt1 mutant) intestines were harvested one week following tamoxifen treatment for gene expression and immunostaining analysis. (**A**) qRT-PCR comparing the relative gene expression levels of Dnmt1, Dnmt3a, and Dnmt3b in the jejunum of tamoxifen-treated controls and Dnmt1 mutants (n=3–4 per genotype). Compared to controls, Dnmt1 mutants express significantly lower levels of Dnmt1, while Dnmt3b expression is significantly increased. Gene expression was calculated relative to the geometric mean of TBP and β-actin. p<0.01, Student's *t*-test. For data and *p*-values, refer to **Figure 3—source data 1**. (**B–C**) Controls (**B**) and Dnmt1 mutants (**C**) display similar levels of Dnmt3a protein (green). (**D–E**) Dnmt1 mutants (**E**) display elevated Dnmt3b protein in intestinal crypts, compared to controls (**E**). Error bars represent S.E.M. Scale bars are 50 μm. For all staining, n=3 biological replicates.

The following source data and figure supplements are available for figure 3:

**Source data 1.** Contains qPCR data and analysis shown in **Figure 3A**.

**Source data 2.** Contains targeted bisulfite sequencing data presented in **Figure 3—figure supplement 2**.

**Figure supplement 1.** Deletion of *Dnmt3a* in addition to *Dnmt1* causes no additive effects on epithelial proliferation, genome stability or cell death within one week.

**Figure supplement 2.** Ablation of *Dnmt3a* and *Dnmt1* induces genome demethylation at *LINE1* and *H19* loci.

significant across the entire region (**Figure 5A**). The H19 imprinting control region was slightly demethylated in compared Dn*mt1;Dnmt3b* mutants to controls, but was only significant when comparing the entire sequenced region (**Figure 5—figure supplement 1**).

Histological examination revealed a grossly abnormal epithelium in *Dnmt1;Dnmt3b* mutantmice, with many areas lacking villi and/or crypts completely (**Figure 5B–C**). We performed immunohistochemistry for Dnmt1 and Dnmt3b to confirm loss of both proteins in the majority of the epithelium (**Figure 5D–G**). Differentiation in the double mutant intestine was largely unaffected, and exhibited normal distribution of goblet cells, Paneth cells, and enterocytes (data not shown). Unlike the situation in Dnmt1 mutants, many crypts in double Dnmt1;Dnmt3b mutants were completely Ki67-negative, and harbored extensive DNA damage as indicated by γH2AX foci (**Figure 5H–I**). TUNEL staining revealed increased crypt cell apoptosis in *Dnmt1;Dnmt3b* mutants compared to sibling controls (**Figure 5J–K**). Overall, the *Dnmt1;Dnmt3b* double mutants displayed increased phenotypic severity compared to *Dnmt1* single mutants, characterized by hypomethylation, DNA damage, and cell death.

To confirm that *Dnmt3b* ablation alone does not replicate the *Dnmt1;Dnmt3b* double mutant phenotype, we also analyzed the intestine of *Dnmt1$^{loxP/+}$;Dnmt3b$^{loxP/loxP}$;Villin-CreERT2* (Dnmt3b mutant) mice. Dnmt3b deletion was confirmed by immunoflourescent staining, and we proceeded to further histological analysis (**Figure 5—figure supplement 2**). Loss of Dnmt3b had no affect on intestinal crypt-villus architecture (**Figure 5—figure supplement 2A–B**), and immunostaining for cell proliferation, DNA damage, and apoptosis was similar to controls (**Figure 5—figure supplement 2E–H**). Furthermore, methylation of both the LINE1 repetitive loci and the *H19* imprinting control

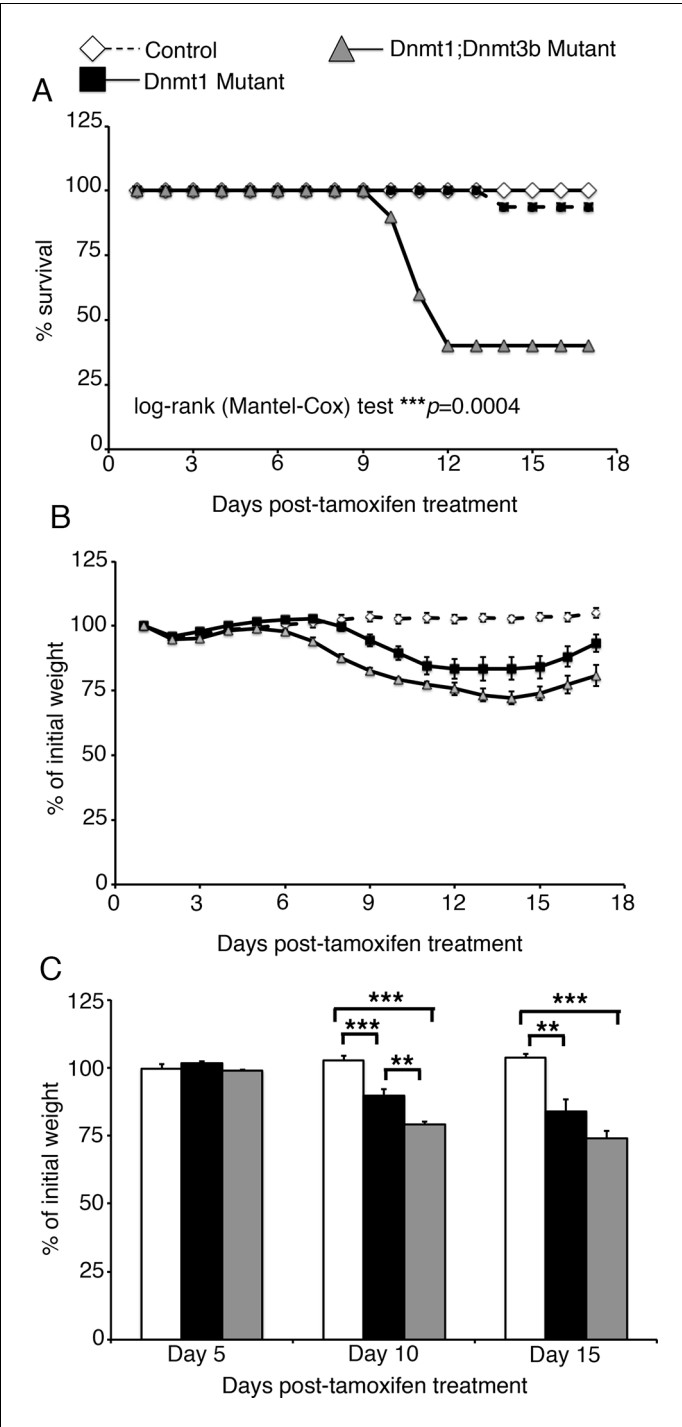

**Figure 4.** Loss of both *Dnmt1* and *Dnmt3b* in the intestinal epithelium results in decreased survival. (**A**) Percent survival of *Dnmt1^{loxP/loxP}*;*Dnmt3b^{loxP/loxP}*(control, n=8), *Dnmt1^{loxP/loxP}*;*Villin-CreERT2* (Dnmt1 mutant, n=16), and *Dnmt1^{loxP/loxP}*;*Dnmt3b^{loxP/loxP}*;*Villin-CreERT2*(Dnmt1;Dnmt3b mutant, n=10). All mice were tamoxifen treated at four weeks of age, and followed 17 days to record weight and survival. Significantly fewer Dnmt1;Dnmt3b mutants survive, compared to both controls and Dnmt1-deficient mice. ***p<0.001, Log-rank test. (**B**) Percent of initial weight each day following tamoxifen treatment in (**A**). Gradual weight loss is observed in both Dnmt1 mutants (white diamonds) and Dnmt1;Dnmt3b mutants (grey triangles). (**C**) Statistical comparison of weight loss between controls, Dnmt1 mutants, and Dnmt1;Dnmt3b mutants. At day 10 post-tamoxifen treatment, both mutant groups have lost a significant amount of weight relative to controls. Dnmt3b;Dnmt1 mutants have also lost significantly

*Figure 4 continued on next page*

*Figure 4 continued*

more weight relative to Dnmt1 mutants. At day 15, both mutant genotypes weigh significantly less than controls. **p<0.01, ***p<0.001, one-way ANOVA. For all data and p-values, refer to *Figure 4—source data 1*.

The following source data and figure supplement are available for figure 4:
**Source data 1.** Contains mouse weight and survival data analysis in *Figure 4*.
**Figure supplement 1.** *Dnmt1;Dnmt3b* mutant intestinal epithelia contain *Dnmt3b*$^+$ escaper crypts, which do not display DNA damage.

region in Dnmt3b mutant crypt cells was equivalent to controls (*Figure 5—figure supplement 3A–B*). These results demonstrate that Dnmt3b alone is not required for intestinal homeostasis.

## Discussion

DNA methylation has been linked to genomic instability in multiple contexts, in both cell lines and in disease. *Dnmt1* hypomorphic mice exhibit increased chromosomal duplications and rearrangements, and develop invasive T-cell lymphomas at approximately four months of age (*Gaudet et al., 2003*). It is important to note that these effects are not restricted to Dnmt1 deficiency. Loss of *Dnmt3b* also induces hypomethylation and chromosomal instability in mouse embryonic fibroblasts (*Dodge et al., 2005*), suggesting that both Dnmt1 and Dnmt3b are crucial for maintaining DNA methylation and preserving genome integrity.

In some cases, de novo methyltransferases are essential for methylation of certain elements or enhancers, and cannot be compensated for by Dnmt1. For example, hematopoietic stem cells (HSCs) require Dnmt3a for normal self-renewal and differentiation processes (*Challen et al., 2012*). Loss of Dnmt3a in HSCs causes demethylation at essential stem cell genes, inducing hyper-proliferation and reducing differentiation rates (*Challen et al., 2012*). Dnmt3b also contributes to silencing of germline genes in somatic cells (*Velasco et al., 2010*), and maintenance methylation in ES cells (*Chen et al., 2003*; *Liang et al., 2002*). Our work adds to this body of evidence that implicates a crucial role for DNA methylation in maintaining genome stability. Given that methylation of LINE elements is significantly reduced, it is tempting to speculate that reactivation of retrotransposition might be a contributing factor to genome instability.

Aberrant DNA methylation and genome instability correlate in a number of gastrointestinal pathologies, including inflammatory bowl disease (IBD) and colitis-associated cancer (*Hartnett and Egan, 2012*). The phenotype presented in our *Dnmt1;Dnmt3b* double mutant mice is reminiscent of mouse models of IBD, in which altered epithelial barrier function leads to increased immune cell recruitment and chronic inflammation in the gastrointestinal tract (*Wirtz and Neurath, 2007*). Indeed, a recent study in zebrafish demonstrated that loss of DNA methylation at the tumor necrosis factor alpha (*tnf-a*) promoter prompted increased *tnf-a* expression in the gut epithelium, leading to elevated apoptosis, barrier dysfunction, and immune cell localization (*Marjoram et al., 2015*). In a chemically induced mouse model of IBD, inhibition of DNA methylation aggravated the inflammatory response, suggesting DNA methylation acts to protect against inflammation and IBD (*Kominsky et al., 2011*). Our data support the hypothesis that DNA methylation supports intestinal epithelial homeostasis and helps to maintain crypt architecture.

Current dogma holds that deletion of Dnmt1 is lethal in all dividing somatic cells (*Liao et al., 2015*); conversely, we find that the rapidly dividing intestinal epithelium can survive without Dnmt1. Following acute loss of *Dnmt1*, *Dnmt3b* expression is induced, and methylation of repetitive elements is restored. However, if *Dnmt3b* is ablated concurrently with *Dnmt1*, restoration of DNA methylation is prevented, resulting in massive DNA damage, and cell death. It is important to note that our data does not provide direct evidence for methyltransferase activity of Dnmt3b at hemimethylated CpG sites, which are targeted by Dnmt1. Indeed, it is also possible that Dnmt3b is compensating for the loss of Dnmt1 via repeated cycles of de novo methylation, in which Dnmt3b remethylates regions that are demethylated following Dnmt1 deletion after each cycle of DNA replication.

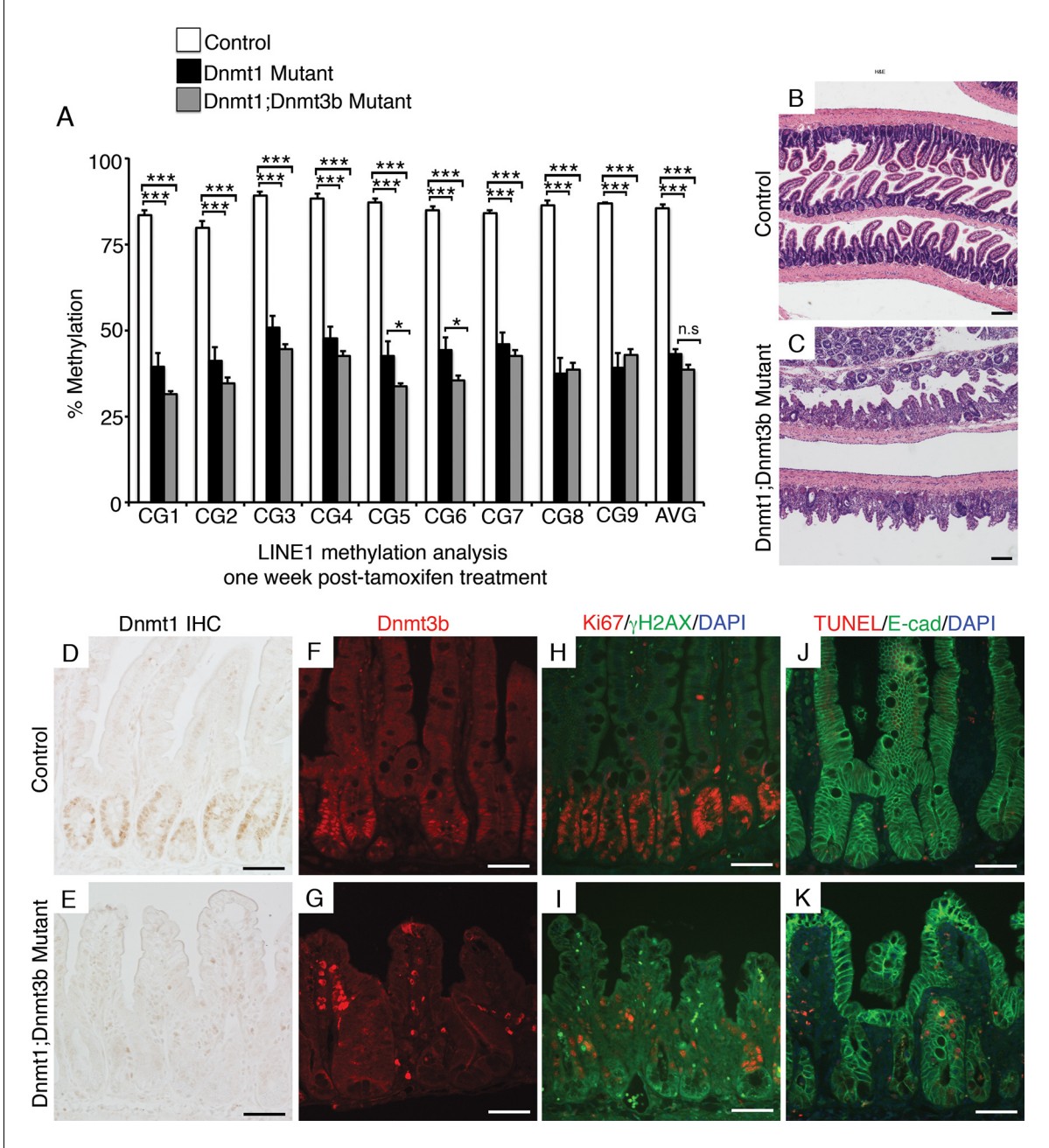

**Figure 5.** Simultaneous loss of *Dnmt1* and *Dnmt3b* results in acute genomic instability, increased apoptosis, and genome demethylation. (A) *Dnmt1^{loxP/loxP}*;*Dnmt3b^{loxP/loxP}*(control), *Dnmt1^{loxP/loxP}*;*Villin-CreERT2* (Dnmt1 mutants) and *Dnmt1^{loxP/loxP}*;*Dnmt3b^{loxP/loxP}*;*Villin-CreERT2* (Dnmt1;Dnmt3b mutant) intestines were harvested one week following tamoxifen treatment for DNA methylation analysis. Crypts were isolated from paraffin-embedded tissue by laser capture microdissection, and the methylation levels of *LINE1* loci were determined by targeted bisulfite sequencing. LINE1 methylation is significantly decreased in both Dnmt1 and Dnmt1;Dnmt3b mutants compared (n=4–6 per genotype). Error bars are S.E.M. **p<0.01, one-way ANOVA. For data and *p*-values per CpG, refer to *Figure 5—source data 1*. (B–C) Hematoxylin and eosin staining of Dnmt1;Dnmt3b mutants (C) compared to controls (B). Double mutants display severe crypt and villus loss compared to controls. (D-G) Immunostaining confirms Dnmt1 and Dnmt3b protein loss in the Dnmt1;Dnmt3b mutant intestinal epithelium (E,G) compared to control (D,F). (H,I) Immunofluorescent staining for Ki67 (red), which marks proliferating cells, and γH2AX (green), which marks DNA double-strand break loci as a marker of genome instability. One week following tamoxifen injection, Dnmt1;Dnmt3b mutants (I) display decreased proliferation and increased DNA damage compared to controls (H). (J,K) Immunofluorescent TUNEL staining (red), which marks apoptotic nuclei, and E-cadherin (green), to outline the intestinal epithelium. One week after tamoxifen treatment, Dnmt1;Dnmt3b mutants (K) display increased crypt cell apoptosis relative to controls (J). All scale bars are 50 µm. For all staining, n=3 biological replicates.

*Figure 5 continued on next page*

*Figure 5 continued*

The following source data and figure supplements are available for figure 5:

**Source data 1.** Contains targeted bisulfite sequencing data shown in *Figure 5A*.

**Source data 2.** Contains targeted bisulfite sequencing data presented in *Figure 5—figure supplement 1*.

**Source data 3.** Contains targeted bisulfite sequencing data shown in *Figure 5—figure supplement 3*.

**Figure supplement 1.** Ablation of *Dnmt3b* and *Dnmt1* induces genome demethylation at *H19* loci.

**Figure supplement 2.** *Dnmt3b* deletion has no effect on epithelial proliferation, genome stability, or cell death within one week.

**Figure supplement 3.** Ablation of *Dnmt3b* alone is not sufficient to induce genome demethylation.

Interestingly, we find that Dnmt3a cannot compensate adequately for the loss of both Dnmt1 and Dnmt3b, nor is Dnmt3a required in the absence of Dnmt1. Indeed, we see that the pressure to maintain Dnmt1 and/or Dnmt3b expression is so high that Dnmt3b$^+$ escaper cells proliferate in excess to recover the intestinal epithelium in a small subset of mutants (*Figure 4—figure supplement 1*). Although the basal levels of Dnmt3a mRNA in the control intestinal epithelium are much higher than those of Dnmt3b or Dnmt1 (*Figure 3A*), Dnmt3a does not appear to be necessary to maintain intestinal homeostasis or DNA methylation patterns. This may be due to a number of factors, such as differing abilities of the methyltransferases to interact with cofactors and epigenetic complexes. Both Dnmt1 and Dnmt3b interact with the polycomb group repression complexes (PRC1 and PRC2), and regulate distinct sites in colorectal cancer development (*Jin et al., 2009*). Furthermore, different methyltransferases are required in distinct ways in certain tissue and cell types. For example, in the hematopoietic system, loss of Dnmt3a leads to elevated proliferation rates and cancer development, while loss of Dnmt3b has little to no effect on hematopoietic stem cell function (*Challen et al., 2012*; *2014*).

In conclusion, we show that Dnmt1 and Dnmt3b cooperate to maintain methylation in the adult mouse intestinal epithelium. Loss of Dnmt1 results in short-term hypomethylation, genomic instability, and apoptosis, followed by recovery over time. Dnmt3b is upregulated in response to deletion of *Dnmt1* in the adult intestine, and is required to recover DNA methylation and epithelial integrity. These results provide the first example of a rapidly dividing somatic tissue that can survive in the absence of Dnmt1, and suggest that the strict division of the Dnmt enzymes into 'de novo' and 'maintenance' methyltransferases might not fully represent the situation in vivo.

## Materials and methods

### Mice

*Dnmt1*$^{loxP/loxP}$ and *Dnmt3b*$^{loxP/loxP}$ mice were provided by Rudolf Jaenisch (*Jackson-Grusby et al., 2001*; *Lin et al., 2006*). *Dnmt3a*$^{loxP/loxP}$ mice were provided by En Li (*Kaneda et al., 2004*), and *Villin-CreERT2* mice were received from Sylvie Robine (*El Marjou et al., 2004*). For *Dnmt1* and *Dnmt3b* deletion experiments, Cre-recombination was induced by three daily intraperitoneal injections of 1.6 mg tamoxifen (Sigma-Aldrich, St. Louis, MO) in an ethanol/sunflower oil mixture. In all experiments, littermate controls without the *Villin-CreERT2* transgene were also tamoxifen treated. All procedures involving mice were conducted in accordance with approved Institutional Animal Care and Use Committee protocols.

### Histology

Tissues were isolated and fixed using 4% paraformaldehyde in PBS and then embedded in paraffin. Antigen retrieval was performed using the 2100 Antigen-Retriever in Buffer A (Electron Microscopy Sciences, Hatfield, PA) and standard immunostaining procedures were performed for Dnmt1 (Santa Cruz), Dnmt3a (Santa Cruz Biotechnology, Dallas, TX), Dnmt3b (Imgenex, San Diego, CA), E-Cadherin (BD Biosciences, San Jose, CA), Ki67 (BD Biosciences), and γH2AX (Cell Signaling Technology, Beverly, MA). TUNEL staining was performed using TUNEL Label and Enzyme (Roche, Indianapolis,

**Table 1.** qRT-PCR primer sequences.

| Gene | Forward 5'-3' | Reverse 5'-3' |
| --- | --- | --- |
| Beta-actin | GAAGTGTGACGTTGACATCCG | GTCAGCAATGCCTGGGTACAT |
| TBP | CCCCTTGTACCCTTCACCAAT | GAAGCTGCGGTACAATTCCAG |
| Dnmt1 | CTTCACCTAGTTCCGTGGCTA | CCCTCTTCCGACTCTTCCTT |
| Dnmt3a | GCACCAGGGAAAGATCATGT | CAATGGAGAGGTCATTGCAG |
| Dnmt3b | GGATGTTCGAGAATGTTGTGG | GTGAGCAGCAGACACCTTGA |

IN) and AlexaFluor 555-aha-dUTP (Molecular Probes, Eugene, OR). All microscopy was performed on a Nikon Eclipse 80i (Tokyo, Japan). For all immunofluorescence and immunohistochemistry staining, n=3 biological replicates per genotype, per timepoint.

## Laser capture microdissection
Crypt cell DNA was collected using a Leica LMD7000 Laser Microdissection microscope (Wetzlar, Germany) and the Arcturus PicoPure DNA isolation kit (Applied Biosystems, Carlsbad, CA).

## qRT-PCR
Intestines were gently scraped to remove villi, and treated with EDTA to isolate crypt cells. RNA was extracted using the Trizol RNA isolation protocol (Invitrogen, Carlsbad, CA), followed by RNA cleanup using the RNeasy Mini Kit (Qiagen, Hilden, Germany). mRNA expression was determined using quantitative RT-PCR, as described previously (*Gupta et al., 2007*). The SYBR green qPCR master mix (Agilent, Santa Clara, CA) was used in all qPCR reactions, and the fold change was calculated relative to the geometric mean of *Tbp* and *β-Actin*, using the △CT method. The method of normalizing to the geometric mean of a set of reference genes has been described previously (*Vandesompele et al., 2002*). Primer sets can be found in *Table 1*.

## Targeted bisulfite sequencing
100 ng of mouse genomic DNA was bisulfite converted using the Epitect bisulfite kit (Qiagen). Template DNA was amplified using KAPA HIFI Uracel+ (KAPA Biosystems, Wilmington, MA) with primers directed to the LINE1 and H19 regions (*Table 2*). Sequencing libraries were prepared and analyzed using the BiSPCR[2] strategy, described previously (*Bernstein et al., 2015*).

## Statistical analysis
Where indicated, GraphPad Prism 6 (La Jolla, CA) was employed to calculate log-rank test and ANOVA statistics.

**Table 2.** Bisulfite sequencing primer sets.

| Gene | Sequence (5'-3') |
| --- | --- |
| H19 PCR#1 Forward | ACACTCTTTCCCTACACGACGCTCTTCCGATCTGTTTGTTGAATTAGTTGTGGGGTTTATA |
| H19 PCR#1 Reverse | GTGACTGGAGTTCAGACGTGTGCTCTTCCGATCTTAAAAAAAAAAACTCAATCAATTACAATCC |
| LINE1 PCR#1 Forward | ACACTCTTTCCCTACACGACGCTCTTCCGATCTGTTAGAGAATTTGATAGTTTTTGGAATAGG |
| LINE1 PCR#1 Reverse | GTGACTGGAGTTCAGACGTGTGCTCTTCCGATCTCCAAAACAAAACCTTTCTCAAACACTATAT |
| Unique Barcode PCR#2 Forward | AATGATACGGCGACCACCGAGATCTACACTCTTTCCCTACACGAC |
| Unique Barcode PCR#2 Reverse | CAAGCAGAAGACGGCATACGAGATCGTGATGTGACTGGAGTTCAGACGTGT |

Red text, Illumina adapter sequence; blue text, Unique Illumina Sequencing Barcodes, 1–48.

## Acknowledgements

We thank Haleigh Zillges and Dr. Jonathan Schug for technical assistance. We acknowledge services provided by the University of Pennsylvania Functional Genomics Core (P30-DK19525). We thank the University of Pennsylvania Department of Dermatology, especially Dr. John Seykora and Dr. Stephen Prouty, for Leica LCM training and equipment access. We acknowledge the support of Dr. Adam Bedenbaugh and the Morphology Core of the Penn Center for the Study of Digestive and Liver Diseases (P30-DK050306).

## Additional information

### Funding

| Funder | Grant reference number | Author |
|---|---|---|
| National Institute of Diabetes and Digestive and Kidney Diseases | R37-DK053839 | Klaus H Kaestner |

The funders had no role in study design, data collection and interpretation, or the decision to submit the work for publication.

### Author contributions

ENE, Planned experiments and prepared the manuscript, Performed experiments, Read, commented on and approved the manuscript, Acquisition of data, Analysis and interpretation of data; KLS, Planned experiments and prepared the manuscript, Performed experiments, Read, commented on and approved the manuscript, Conception and design, Acquisition of data, Analysis and interpretation of data; KHK, Conception and design, Analysis and interpretation of data, Drafting or revising the article

### Author ORCIDs

Klaus H Kaestner, http://orcid.org/0000-0002-1228-021X

### Ethics

Animal experimentation: This study was performed in strict accordance with the recommendations in the Guide for the Care and Use of Laboratory Animals of the National Institutes of Health. All of the animals were handled according to approved institutional animal care and use committee (IACUC) protocol (804436) of the University of Pennsylvania. All surgery was performed under isofluranel anesthesia, and every effort was made to minimize suffering.

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
