## [Decision Letter]

Thank you for resubmitting your work entitled "The 'de novo' DNA methyltransferase *Dnmt3b* compensates the *Dnmt1*-deficient intestinal epithelium" for further consideration at *eLife*. Your manuscript has been favorably evaluated by Janet Rossant (Senior editor), a Reviewing editor, and two reviewers.

The reviewers have discussed the reviews with one another and the Reviewing editor has drafted this decision to help you prepare a revised submission.

Reviewer #1:

In this manuscript by Eliott et al., the authors report their analysis on the important cooperation of *Dnmt1* and *Dnmt3b* in the maintenance of intestinal epithelial cell homeostasis. Their experiments led to the demonstration of the dramatic rescue, by *Dnmt3b*, of the phenotypes, hypomethylation and genomic instability caused by the loss of *Dnmt1* in the intestinal epithelium. In general, this is a very interesting and well-written manuscript. The results are clearly presented and provide important new information to this particular field. All experiments shown here support the conclusion drawn by the authors.

Reviewer #2:

In the manuscript entitled "The 'de novo' DNA methyltransferase *Dnmt3b* compensates the *Dnmt1*-deficient intestinal epithelium", Elliott et al. examine the role of DNA methylation in mammalian epithelial homeostasis by investigating the effect of *Dnmt1, Dnmt3a* and *Dnmt3b* knockout in murine intestine. Using appropriate genetic methods, they proved that *Dnmt3b*, but not *Dnmt3a*, can replace or at least compensate for *Dnmt1* to keep the integrity of the gut epithelia. Thus, this manuscript presents solid genetic evidence that these enzymes are functionally redundant in the gut epithelium. Considering the popular dogmatic view that Dnmt enzymes can be divided into two classes: 'de novo' and 'maintenance' methyltransferases, this novel finding is very important and I therefore recommend its publication in *eLife*. There are few additional points that the authors may discuss further:

1) It seems appropriate to specify the region where the authors have observed global hypomethylation at the end of the subsection “*Dnmt1* ablation results causes weight loss, hypomethylation, and genomic instability “by adding 'in LINE1 repeats and H19 imprinting region'.

2) *Dnmt3a* and *Dnmt3b* are closely related methyltransferases both in sequence homology and domain architecture. It is therefore very interesting that *Dnmt3a* couldn't compensate the loss of *Dnmt1* even though the basal level of *Dnmt3a* is higher than that of *Dnmt3b*. The authors may discuss this interesting point in the Discussion.

3) *Dnmt1;Dnmt3b* compound mutants showed severe phenotype and more than 50% lethality. However, after two weeks, the phenotype of these animals stabilized with regards to both survival rate and weight loss. Does this mean that there are additional methyltransferases that can further compensate for the loss of the two enzymes? Do the authors have any idea which methyltranserases could be involved in this compensatory mechanism? Can the authors perform for example qPCR analysis to further investigate this?

4) The number of goblet cells seems to be altered in the *Dnmt1;Dnmt3b* compound mutant intestine. It is recommended to provide higher magnification images of H&E and staining for markers of goblet cells (Alcian blue), Paneth cells (Lysozyme), enteroendocrine cell (Chromogranin) and intestinal stem cells (Olfm4/Lgr5). Together this data will provide basic information of the phenotype of this precious compound mutant mouse.

---

## [Author Response]

Reviewer #2: In the manuscript entitled "The 'de novo' DNA methyltransferase Dnmt3b compensates the Dnmt1-deficient intestinal epithelium", Elliott et al. examine the role of DNA methylation in mammalian epithelial homeostasis by investigating the effect of Dnmt1, Dnmt3a and Dnmt3b knockout in murine intestine. Using appropriate genetic methods, they proved that Dnmt3b, but not Dnmt3a, can replace or at least compensate for Dnmt1 to keep the integrity of the gut epithelia. Thus, this manuscript presents solid genetic evidence that these enzymes are functionally redundant in the gut epithelium. Considering the popular dogmatic view that Dnmt enzymes can be divided into two classes: 'de novo' and 'maintenance' methyltransferases, this novel finding is very important and I therefore recommend its publication in eLife. There are few additional points that the authors may discuss further: 1) It seems appropriate to specify the region where the authors have observed global hypomethylation at the end of the subsection “Dnmt1 ablation results causes weight loss, hypomethylation, and genomic instability “by adding 'in LINE1 repeats and H19 imprinting region'.

We altered the text, to read “Overall, these results suggest a phenotype in which loss of *Dnmt1* results in hypomethylation of LINE1 repeats and the H19 imprinting control region, increased DNA damage, and apoptosis.”

*2) Dnmt3a and Dnmt3b are closely related methyltransferases both in sequence homology and domain architecture. It is therefore very interesting that Dnmt3a couldn't compensate the loss of Dnmt1 even though the basal level of Dnmt3a is higher than that of Dnmt3b. The authors may discuss this interesting point in the Discussion.*

Thank you for this helpful suggestion. We added a new paragraph to the Discussion to review the different functions of de novo methyltransferases, and to examine why *Dnmt3a* cannot compensate for *Dnmt1* and *Dnmt3b* loss.

3) Dnmt1;Dnmt3b compound mutants showed severe phenotype and more than 50% lethality. However, after two weeks, the phenotype of these animals stabilized with regards to both survival rate and weight loss. Does this mean that there are additional methyltransferases that can further compensate for the loss of the two enzymes? Do the authors have any idea which methyltranserases could be involved in this compensatory mechanism? Can the authors perform for example qPCR analysis to further investigate this?

Reviewer #2was concerned about *Dnmt1;Dnmt3b* double mutants that were able to survive, and wanted us to provide evidence of a compensatory mechanism. Thank you for this comment, as we believe answering this question in more detail adds to the quality of our manuscript. As mentioned in the text, we believe that *Dnmt3b*-positive cells, which escape Cre-loxP recombination, proliferate and repopulate the intestinal epithelium. We have provided a new figure, Figure 4—figure supplement 1, which shows an example of a *Dnmt3b*-positive escaper crypt in the double mutant small intestine. We show that this escaper crypt does not display the DNA damage phenotype present in mutant crypts by co-staining for *Dnmt3b* and γH2AX. γH2AX is only present in the mutant crypts, and is not present in the *Dnmt3b*-positive escaper crypts.

*4) The number of goblet cells seems to be altered in the Dnmt1;Dnmt3b compound mutant intestine. It is recommended to provide higher magnification images of H&E and staining for markers of goblet cells (Alcian blue), Paneth cells (Lysozyme), enteroendocrine cell (Chromogranin) and intestinal stem cells (Olfm4/Lgr5). Together this data will provide basic information of the phenotype of this precious compound mutant mouse.*

Reviewer #2’sfinal concern was that we should provide further characterization of the differentiation of the double mutant intestinal epithelium. He suggested staining for goblet cells, Paneth cells, enterocytes, and Olfm4 and Lgr5 to mark intestinal stem cells. This amount of staining, in multiple biological replicates, would take several months to complete, and would probably not yield data to further the main point of our manuscript. Furthermore, Olfm4 and Lgr5 would require sensitive in situ hybridization, because antibodies are not readily available. We have preliminary evidence to suggest that differentiation (goblet cells, Paneth cells, enterocytes) is largely unaffected in these double mutants, which we have indicated in the text.